# Toxicity and Anti-Inflammatory Activity of Phenolic-Rich Extract from *Nopalea cochenillifera* (Cactaceae): A Preclinical Study on the Prevention of Inflammatory Bowel Diseases

**DOI:** 10.3390/plants12030594

**Published:** 2023-01-29

**Authors:** Emanuella de Aragão Tavares, Gerlane Coelho Bernardo Guerra, Nadja Maria da Costa Melo, Renato Dantas-Medeiros, Elaine Cristine Souza da Silva, Anderson Wilbur Lopes Andrade, Daline Fernandes de Souza Araújo, Valéria Costa da Silva, Ana Caroline Zanatta, Thaís Gomes de Carvalho, Aurigena Antunes de Araújo, Raimundo Fernandes de Araújo-Júnior, Silvana Maria Zucolotto

**Affiliations:** 1Graduate Program in Drug Development and Technological Innovation, Federal University of Rio Grande do Norte, Natal 59078-970, Brazil; 2Department of Biophysics and Pharmacology, Biosciences Center, Federal University of Rio Grande do Norte, Natal 59078-970, Brazil; 3Graduate Program in Pharmaceutical Science, Federal University of Rio Grande do Norte (UFRN), Natal 59078-970, Brazil; 4Health Sciences College of Trairi, Federal University of Rio Grande do Norte, Santa Cruz 59078-970, Brazil; 5Department of Biomolecular Sciences, Faculty of Pharmaceutical Sciences of Ribeirão Preto, São Paulo University, São Paulo, Ribeirão Preto 14040-903, Brazil; 6Program Degree in Health Science, Federal University of Rio Grande do Norte (UFRN), Natal 59078-970, Brazil; 7Cancer and Inflammation Research Laboratory, Morphology Department, Biosciences Center, Federal University of Rio Grande do Norte, Natal 59078-970, Brazil

**Keywords:** Cactaceae, functional foods, herbal medicines, flavonoids, inflammatory chronic diseases, colitis

## Abstract

Phenolic compounds have been scientifically recognized as beneficial to intestinal health. The cactus *Nopalea cochenillifera*, used as anti-inflammatory in traditional medicine, is a rich source of these bioactive compounds. The present study aimed to investigate the phytochemical profile of *N. cochenillifera* extract and evaluate its acute toxicity and anti-inflammatory effect on 2,4-dinitrobenzenesulfonic acid (DNBS)-induced colitis in rats. The total phenolic content per gram of dry extract was 67.85 mg. Through HPLC-IES-MS^n^, a total of 25 compounds such as saccharides, organic acids, phenolic acids and flavonoids were characterized. The dose of 2000 mg/kg of extract by an oral route showed no signs of toxicity, mortality or significant changes in biochemical and hematological parameters. Regarding intestinal anti-inflammatory effects, animals were treated with three different doses of extract or sulfasalazine. Macroscopic analysis of the colon indicated that the extract decreased the disease activity index. Levels of IL-1β and TNF-α decreased, IL-10 increased and MDA and MPO enzyme levels decreased when compared with the control group. In addition, a down-regulation of MAPK1/ERK2 and NF-κB p65 pathway markers in colon tissue was observed. The epithelial integrity was improved according to histopathological and immunohistological analysis. Thus, the extract provided strong preclinical evidence of being effective in maintaining the remission of colitis.

## 1. Introduction

Inflammatory bowel disease (IBD) is an exponentially growing disease worldwide, especially in developed countries [1,2]. The two most important pathologies comprised in this spectrum are Crohn’s disease (CD) and ulcerative colitis (UC). The pathogenesis of intestinal inflammation is associated with a combination of genetic and environmental factors that promote an exacerbated immune response, resulting in a continuous and progressive inflammation [3].

The current trend in treatments for IBD patients involves the use of medicines associated with complementary therapies, probably as a consequence of the lack of efficacy of conventional treatment or due to their side effects [4,5,6]. The combination of botanical extracts and approved medicines by IBD patients as an add-on therapy has shown a clinical improvement in the quality life of patients through a decrease in the disease activity index and the maintenance of remissive disease status [7].

In this sense, phenolic compounds have showed a great potential to act in the prevention and treatment of IBD, due to their antioxidant and anti-inflammatory potential [8,9] as well as their low toxicity and side effects [10]. Its remarkable anti-inflammatory and antioxidant capacity is due to multiple targets of action such as the inhibition of the production or the action of pro-inflammatory mediators and the gene expression, while synthetic drugs generally act in a single way [11,12,13]. Some clinical studies have confirmed the effectiveness of combinations of approved medicines and supplementations with polyphenols in inducing clinical and endoscopic remission in patients with IBDs [14,15,16].

In the state of science, phenolic compounds, in turn, are known for their antiproliferative activity, regulating the cell DNA repair system and positively stimulating junction proteins to ensure the integrity of the intestinal epithelial mucosa [17]. In addition, they can effectively decrease the inflammatory response of the cyclooxygenase-2 (COX-2) and nitric oxide synthase (iNOS) pathways through the regulation of nuclear factor kappa B (NF-κB), Adenosine Monophosphate Activated Protein Kinase (AMPK) and toll-like receptors (TLR) [18,19]. In short, phenolic compounds may play a key role in modulating the colonic microbiota in healthy and pathological conditions [20,21,22].

In this scenario, the cactus *Nopalea cochenillifera* Salm-Dyck is a good source of raw material rich in phenolic compounds. It is native from Mexico, but currently it is widely distributed in the northeastern region of Brazil [23]. Their cladodes are rich in polysaccharides and phenolic compounds [24,25,26], and they have been employed for agricultural, food and medicinal purposes [27]. In traditional medicine, this species is used to treat inflammatory diseases and decrease the cholesterol level, blood pressure, kidney problems, urinary problems and diabetes [28,29]. Previous studies have reported that this plant showed potential antibiotic and antifungal activities in vitro assays [24,25] and decreased the blood glucose level in an in vivo study [30] and in a pilot clinical trial [31]. Regarding the anti-inflammatory effect, the oral administration of the hydroethanolic extract of *N. cochenillifera* cladodes showed significant anti-inflammatory activity in the rat granuloma induction method [32].

It is worth mentioning that *N. cochenillifera* represents a good plant choice for cultivation in the Northeast region of Brazil due its resistance in drought season, its management being easy and because it can assist in the sustainable development of the region through the establishment of the entire productive chain to obtain raw material to be applied in pharmaceuticals or foods [23].

Hence, the aim of this study was to investigate the phytochemical content of *N. cochenillifera* hydroethanolic extract (NCHE), evaluate its acute toxicity in an in vivo model and investigate the preventive effect of nopal extract in a colitis experimental model of intestinal inflammation induced by 2.4-dinitrobenzene sulfonic (DNBS) in rats. In this study, its protective actions against oxidative and inflammatory stress were studied, and the macroscopic and microscopic changes in the colon after treatment were evaluated. Finally, lipid peroxidation, cytokine dosage and gene expression of the colon samples were also analyzed.

## 2. Results

### 2.1. Physicochemical Analysis of Nopalea cochenillifera Extract

Table 1 shows the results of physicochemical analysis (pH, molar acidity, moisture, ashes, ether extract, crude fiber, protein and total carbohydrates) found in NCHE. It is worth mentioning that NCHE presents a high content of carbohydrates (67.863 ± 0.00), and phenolics (67.85 ± 0.04) and a low content of lipids.

The phytochemical analysis of NCHE through mass spectrometry (HPLC-IES-MS^n^) in positive, and negative ionization modes allowed the characterization of saccharides, organic acids, phenolic acids and flavonoids. The compounds were characterized based on assignments of their mass spectra, using the precursor ion, fragment ions and comparison of the fragmentation patterns with molecules provided by the GNPS library and described in the literature. The putative identification of these compounds is summarized in Table 2. Each compound was assigned a number following the sequence of its retention time. A complete set of extracted ion chromatograms is given in (Appendix A).

### 2.2. Acute Toxicity

In the acute toxicity test, no mortality was observed in the experimental group that received an oral administration of NCHE at a dose of 2000 mg/kg for a short period of 48 h and a prolonged period of 14 days. The animals survived until the end of the observation period. Body weight and feed consumption did not change during the 14-day period when compared to the control group (Figure 1).

Table 3 present the results of the biochemical and hematological tests, respectively. NCHE administration induced no treatment-related adverse effects with regard to body weight, general behavior, relative organ weights and hematological and biochemical parameters of the animals in the treated group when compared to the control group. For comparison purposes, the specialized literature recommends the standardization of values proper for healthy animals of each vivarium/laboratory respecting the local characteristics [46,47].

No changes were observed between groups in the Hippocratic screening. In the open field test, there was no significant difference in ambulation, rearing behavior and grooming activities when comparing the control group to those treated with NCHE. On day 7, there was an increase in the frequency of standing up in the group treated with the 2000 mg/kg dose of NCHE. A decrease in defecation was also observed in the control group during the experiment (Table 4).

In the evaluation of the motor activity of the animals by means of the dwell time (seconds) on the rotating bar of the rotarod apparatus, no significant differences were recorded between the control group and the treated animals. The data are presented in Table 4.

However, the organs showed no abnormal macroscopic changes, and the relative weights of the organs had no statistically significant difference between the animals treated with NCHE and saline; thus, the relative weights of the liver, spleen, kidneys, heart and lung were not affected with the administration of NCHE at the dose of 2000 mg/kg (Table 5). There were also no external morphological changes in the texture or color of these organs when compared to the control group.

In the microscopic analysis of the organs (liver, kidneys and spleen) performed with the control group and with the animals treated with NCHE (2000 mg/kg), no abnormalities were found, and the organs analyzed presented normal architecture, suggesting that there were no morphological changes (Figure 2).

### 2.3. Effect of Nopalea cochenillifera Extract on DNBS-Induced Colitis in Rats

#### 2.3.1. The Disease Activity Index (DAI), Macroscopic Score and Weight/Colonic Length Ratio

The DNBS group showed important characteristics related to IBD, evidenced throughout the experiment, such as a loss of body weight (Figure 3C), presence of blood in the perianal region and altered stool consistency, represented in DAI parameters evaluated in the DAI (Figure 3D).

The macroscopic colonic damage observed in the DNBS control group was characterized by large areas of tissue damage, with the presence of ulcerations and thickening of the intestinal wall (Figure 3B) leading to a mean macroscopic damage score of 9.0 (Figure 3F), also represented by the weight/length of the intestine.

#### 2.3.2. Effect of *Nopalea cochenillifera* Extract on Malondialdehyde Levels and Myeloperoxidase Activity

The amount of malondialdehyde (MDA) levels significantly decreased in the colonic tissue samples of the SSZ group and NCHE-treated rats at doses of 100, 200 and 300 mg/kg when compared to the DNBS group, as can be seen in Figure 4A. To verify whether treatment with NCHE might possibly cause an effect on reducing neutrophil infiltration into the intestinal mucosal layer, myeloperoxidase (MPO) activity was measured in the colonic tissue. As shown in Figure 4B, the colitis induced significant recruitment of intestinal neutrophils. These parameters were significantly reduced in the SSZ group and the NCHE-treated groups at doses of 200 and 300 mg/kg.

#### 2.3.3. Effect of the *Nopalea cochenillifera* Extract on the Colonic Levels of TNF-α, IL-1β, and IL-10

TNF-α levels were significantly increased in the DNBS control group and remained at baseline levels in the healthy group. On the other hand, it reduced in the SSZ group (*p* < 0.05) and in all groups treated with NCHE, compared to the DNBS group. Additionally, the pre-treatment revealed that NCHE at doses of 200 and 300 mg/kg significantly reduced IL-1β levels. However, NCHE at a dose of 100 mg/kg did not differ statistically from the DNBS control. IL-10 levels were significantly increased in the SSZ group and in the treatment with NCHE at doses of 200 and 300 mg/kg compared to the DNBS group, and in NCHE (100 mg/kg), no significant increase was observed (*p* < 0.05) (Figure 5).

#### 2.3.4. Effect of the *Nopalea cochenillifera* Extract on the Gene Expression of MAPK1, NF-κB p65, MUC-2 and ZO-1

The quantitative real-time polymerase chain reaction (qPCR) experiment for the DNBS group exhibited an increased mRNA expression of MAPK1 (ERK2) and NF-κB-p65 when compared to the healthy group. In addition, DNBS-induced colonic damage showed decreases in gene expression of MUC-1 and ZO-1 when compared to the healthy group. The SSZ and NCHE groups decreased the mRNA expression of MAPK1/ERK2 and NF-κB p65. The data demonstrated increased mRNA expression for MUC-2 and ZO-1 in the groups treated with SSZ and NCHE at doses of 200 and 300 mg/kg (*p* < 0.05) compared to the DNBS group (Figure 6).

#### 2.3.5. Histological and Immunohistochemical Analysis

Histological analysis shows a statistically significant difference in histopathological scores for the SSZ group (*p* < 0.05; Figure 7G) and in all groups treated with NCHE compared to the DNBS group. This result suggests that NCHE treatment contributes to decreased inflammation and regeneration of the intestinal mucosa. 

In the DNBS group, the colonic tissue showed intense inflammation characterized by areas of intense inflammatory infiltrate of polymorphonuclear cells, in addition to an area of ulceration of the intestinal mucosa with a loss of integrity of the basal membrane (Figure 7B). The SSZ group shows the intestinal mucosa in the process of regeneration and a moderate inflammatory infiltrate, and a similar result was observed in the groups treated with NCHE at doses of 200 and 300 mg/kg (Figure 7E,F, respectively).

The immunohistochemical staining of NF-κB p65 and COX-2 in colonic tissue samples is shown in Figure 8. In the healthy group, the expression of NF-κB-p65- and COX-2-positive cells were not observed. However, the administration of DNBS caused an increase in the expression of NF-κB p65 and COX-2 in the colon tissue compared to the healthy group (Figure 8). In contrast, the sulfasalazine and NCHE at doses of 200 and 300 mg/kg reduced the expression of NF-κB p65- and COX-2-positive cells compared to the DNBS group (*p* < 0.05; Figure 8).

## 3. Discussion

### 3.1. The Physicochemical and Phytochemical Characterization of Nopalea cochenillifera Extract Identifies the Presence of Compounds Beneficial to Intestinal Health

The physicochemical characterization of the NCHE showed a significant percentage of crude fiber and a high amount of carbohydrates and phenolic compounds. These findings are in accordance with a previous study reported by Silva et al. (2015) [48]. Lima et al. (2021) [49] also reported the high content of total carbohydrates (66.80% of fresh raw material-cladodes) in *N. cochenillifera* ethanolic extract, harvested in Paraíba state, Brazil.

The TPC and TFC in NCHE was expressed in gallic acid and quercetin equivalent (mg) per gram of dry extract with a mean value equal to 67.85 mg GAE/g and 46.16 mg QE/g of dry extract, respectively (Table 2). These values were higher than those reported previously by Alves et al. (2017) [50], who evaluated the TPC and TFC in *Opuntia* and *Nopalea* extracts (95% ethanol) as 5.41 mg GAE/g and 2.7 mg QE/g, respectively, from samples harvested in different cultivars in the rainy season. Fabela-Illescas et al. (2022) [26] found a content of 6.47 mg polyphenol equivalent/g gallic acid and 2.08 mg rutin equivalent/g flavonoid in the cladodes flour of *N. cochenillifera*. Then, we can suggest that 70% ethanol is a good choice of solvent to improve the extraction of phenolic compounds. In previous work, our research group optimized the best percentage of ethanol to increase the extraction efficiency of phenolic compounds in *Moringa oleifera* and proved that harvesting in the drought period (typical season from the northeast region of Brazil) is crucial to increase the phenolic content in raw material [51]. TPC and TFC per gram of NCHE is an essential quality control parameter related to the pharmacological preclinical effect found in this work that ensures the reproducibility of raw material [52].

HPLC-ESI-MS^n^ analysis allowed a characterization of the phytochemical profile of NCHE. The dereplication of the negative and positive HPLC-ESI-MS^n^ data, in combination with GNPS public libraries, literature information and the fragmentation pattern study (MS^2^ and MS^3^ experiments), allowed us to annotate 25 metabolites present in NCHE (Table 3). The sugar portion of the *O*-glycosylated compounds was determined by analysis of fragmentation pathways of the precursor ions in the positive and negative mode, such as the cleavage of saccharide residues, leading to characteristic neutral losses of 132 (pentose), 146 (deoxyhexose) and 162 Da (hexose) [53,54]. MS/MS spectra analysis of the precursor ions [M+H]^+^ at m/z 757 (**14**), 771 (**16**), 611 (**18**), 595 (**22**) and 625 (**23**) showed characteristic fragmentations of the flavonoids *O*-glycosylated with sequential losses of deoxyhexose and hexose moiety, which led to the formation of the product ions Y_0_^+^ at m/z 287 and m/z 303, found to be associated with the flavonol aglycones of kaempferol and quercetin, respectively. In addition, second-order fragmentation for compounds **16** and **23** resulted in neutral losses of 15 and 32 Da, corresponding to ^•^CH_3_ and CH_3_OH, respectively [55,56].

The precursor ions [M+H]^+^ at *m/z* 447 (**19**), 461 (**20**) and 547 (**24** and **25**) showed a fragmentation pattern similar to flavonoids due to the loss of sugar residues, which resulted in the formation of Y_0_^+^ ions at *m*/*z* 285 (**19**) and at *m/z* 299 (**20**, **24** and **25**) and the radical elimination of methyl groups. This fragmentation pattern, which is in agreement with reports published in the literature, implies the presence of methoxylated isoflavones [42]. The presence of isoflavones is almost entirely restricted to the Fabaceae subfamily of the Leguminosae family, but they are also occasionally found in some other families [57]. The second-order fragmentation of compounds **24** and **25** resulted in a characteristic loss of 248 Da [M+H−86−162]^+^, indicating an acylation at the glycosyl residue, which was assigned to the malonyl hexose group [53]. The analysis of the MS/MS mass spectra recorded for the precursor ions at m/z 476 (**15**), 562 (**17**) and 314 (**21**) showed representative fragmentation patterns of the amides of hydroxycinnamic. The second-order mass spectra of these ions showed similar characteristic losses of 137 Da and the fragment at *m*/*z* 177, corresponding to the tyramine and the ferulic acid residue, respectively [43,58]. Other organic compounds were also observed, including several organic acids (compounds **2**, **5** and **6**), benzoic acids (compounds **8**–**9** and **13**) and hydroxycinnamic acid and their derivatives (compounds **10**–**12**). (Table 3). Partial results corroborate with Fabella-Illescas et al. (2022) [26], who identified and quantified the first-time phenolic compounds such as acid gallic, acid ferulic, acid chlorogenic, acid *p*-coumaric, acid siring and acid neochlorogenic in samples of *N. cochenillifera* cladodes flour through HPLC, but to the best of our knowledge, this is the first dereplication study and description of isoflavones for this species.

Taken together, it is possible to associate the anti-colitis effect observed in preclinical studies with the phytochemical content found in NCHE. To strengthen our findings, clinical studies about supplementation with flavonoids in patients with IBD have demonstrated a positive correlation in the maintenance of remissive disease status [59,60]. 

### 3.2. Nopalea cochenillifera Hydroethanolic Extract did Not Produce Toxic Effects in Rats after Acute Treatment

In toxicity studies, changes in organ weight are sensitive indicators of toxicity and physiological disturbances [61,62]. In toxicity studies, it is possible to infer that the administration of NCHE at a single dose of 2000 mg/kg did not interfere in the spontaneous locomotor activity and anxiety of the animals, as it did not change the parameters of total distance covered, lifting and grooming, despite the increase in the number of defecations. Similarly, no alterations were observed in motor coordination, due to non-interference in the time spent on the rotating bar on all evaluated days, using the rotarod test.

The analysis of hematological parameters is widely used to assess the possible degree of toxicity of drug substances and botanical extracts [63]. Changes in the hematopoietic system have a higher predictive value for human toxicity when data are translated from animal studies [64]. Changes in platelet numbers can also point to a homeostatic imbalance and are indicative of thromboembolic problems [65]. 

The acute in vivo toxicity study was the first report for this species, and it is considered important safety data for further studies. The NCHE can be considered of low acute toxicity according to the class method recommended by OECD 423 (OECD, 2001) [66] and fits in Type 5 (substance with LD50 higher than 2000 mg/kg and less than 5000 mg/kg), because there were no signs of general toxicity such as changes in water and feed consumption, body weight, hematological and biochemical parameters, and there were no behavioral changes and no deaths observed during the whole experimental period. Furthermore, the extract did not cause morphological changes in the organs investigated microscopically.

### 3.3. Effects of Nopalea cochenillifera Hydroethanolic Extract on Inflammatory Markers, Oxidative Stress and Intestinal Permeability in Rats with Induced Colitis

Based on the preclinical safety results, this study followed the evaluation of NCHE anti-inflammatory effect since this species is a good raw material source of fibers, polysaccharides and polyphenols [26]. For these compounds, there is attributed an anti-inflammatory and antioxidant effect that can justify their action to suppress intestinal inflammation, both in preclinical and clinical studies [67,68]. So, in this study, we have chosen to investigate the anti-colitis effect of *N. cochenillifera* extract. Again, to the best of our knowledge, this is the first report that associates *N. cochenillifera* with inflammatory bowel diseases.

The in vivo model of DNBS induction of intestinal inflammation causes severe acute inflammation in the colon that reproduces clinical features and an inflammatory response related to Crohn’s disease in humans and generates symptoms such as decreased body weight, diarrhea and blood in the stool, which are parameters considered to assess the DAI in animals [69]. Treatment with sulfasalazine and NCHE promoted significantly lower DAI and weight loss on all days after colitis induction. A significant loss in body weight was observed in the colitis groups, indicating severe colitis progression; however, the treatments ameliorated the weight loss. The NCHE also showed a decrease in the ratio of colonic weight (mg) to length (cm), an indirect indicator of edema and inflammation. The beneficial effect of NCHE was accompanied by a reduction in macroscopic damage. IBD’s (CD and UC) are characterized by the overactivation of the intestinal immune system, which results in chronic inflammation, with the release of inflammatory cytokines, thus exacerbating tissue damage. Infiltration of immune cells, particularly neutrophils, is a histopathological feature of IBD. Neutrophils and monocytes are recruited to the intestinal mucosa and enhance the inflammatory response by producing inflammatory cytokines and reactive oxygen species (ROS). Both neutrophils and monocytes secrete the enzyme heme MPO [70]. These biomarkers of oxidative damage, such as MPO, are increased in the mucosa of IBD patients, and their decrease reflects a reduction in inflammation of the injured tissue. This study showed that the animals treated with NCHE showed a statistically significant decrease in MPO activity (Figure 4B), thus corroborating with the histological analyses, which showed a reduced infiltration of inflammatory cells. 

In experimental colitis models as in IBD in humans, the intestinal inflammatory process leading to the induction of oxidative stress was associated with a significant increase in MDA in the intestinal tissue. MDA is a product of lipid peroxidation and contributes to the inflammatory reaction by the activation of pro-inflammatory cytokines [71]. The beneficial effect of NCHE in improving intestinal inflammation induced by DNBS may be associated with its antioxidant properties, in particular the presence of phenolic compounds. It has been shown that polyphenols can act as radical scavenging compounds and are therefore capable of acting as antioxidants [72].

In IBD, there is an increase in inflammatory mediators responsible for generating and maintaining inflammation in the gastrointestinal tract, including pro-inflammatory cytokines such as interleukin IL-1β, IL-6, IL-17, tumor necrosis factor (TNF-α), prostaglandins and nitric oxide (NO) [73]. In this way, polyphenolic compounds also exert anti-inflammatory effects through their ability to decrease the expression of pro-inflammatory molecules, mainly IL-1β, IL-6, TNF-α and COX-2, which play an important role in IBD [74]. Additionally, NCHE was able to increase the levels of IL-10, an anti-inflammatory cytokine, which can be used as a marker of the evolution of intestinal inflammation, given that its levels seem to increase only at the time of disease resolution [75,76]. Nevertheless, we were able to observe an increase in IL-10 at doses of 200 and 300 mg/kg of the extract shortly after the period of disease activation by DNBS (3 days). 

An important mechanism that could justify the beneficial effects of NCHE in experimental models of DNBS-induced colitis is the modulation of the NF-κB p65/MAPK signaling pathway. NF-κB p65 is a key macrophage transcription factor and induces many inflammatory genes, including IL-1β, TNF-α and COX-2. DNBS activates the NF-κB signaling pathway and results in increased expression of the inflammatory cytokines IL-1β and TNF-α, which contributes to the progression of intestinal inflammation [77]. The inflammatory cytokines can trigger the pathway of MAPKs, and the extracellular regulated kinases (ERK1/2), p38 MAPK and JNK represent the three primary MAPK signaling pathways [78]. The NF-κB and MAPKs pathways are closely linked with IBD [79], and therefore, NCHE may improve inflammation for DNBS-induced colitis by modulating both pathways.

The downregulation of the p65 NF-κB pathway negatively regulates the expression of genes associated with the transcription of mediators such as pro-inflammatory cytokines. The decrease in the colonic levels of pro-inflammatory cytokines (IL-1β and TNF-α) analyzed in this study can be attributed to the presence of compounds derived from kaempferol, quercetin and ferulic acid and other phenolic compounds identified in NCHE which have the ability to inhibit toll-like receptors and NF-κB activation [80,81,82]. The gallic acid produced an increase in IκB in human gastric carcinoma cells, thus inhibiting NF-κB activity and the pro-inflammatory cascade [83]. Isoflavones also promoted the inhibition of NF-κB in colon tissues of mice in experimental colitis induced by dextran sulfate sodium [84]. So, we can hypothesize that bioactive compounds from NCHE can act in the same pathway. In this study, a decrease in gene and protein expression of NF-kB is evidenced in the SSZ and NCHE. The mitogen-activated protein kinases (MAPK) signaling is the other important extracellular signal transduction pathway stimulated by inflammatory mediators [78,79].

In fact, the DNBS-induced colitis model in rodents alters the integrity and functionality of intestinal cells, promoting a rupture in the barrier of tight epithelial junctions and imbalances in paracellular permeability. In fact, the intestinal epithelial barrier markers (MUC-2 and ZO-1) showed a lower gene expression in the DNBS group, and on the other hand, SSZ and NCHE promoted a higher expression of these markers. MUC-2 plays an important role in protecting the intestinal mucosa, and its dysfunction or reduced expression contributes to the pathogenesis of IBD, as it directly interferes with protection against pathogenic bacterial invasion and colonization [85]. ZO-1 is part of a subgroup of proteins that make up the tight junctions of the intestinal epithelium, and the increase in its expression is an important factor in reducing the intestinal inflammatory process in IBD [86]. A study with a human colorectal adenocarcinoma cell line suggested that polyphenols can modulate the expression of MUC-2 and ZO-1, protecting the epithelial barrier function of cells exposed to a chemical agent [87].

In summary, the results obtained in this study demonstrate that NCHE can modulate NF-κB p65/MAPK signaling pathways, reduce oxidative stress and inhibit the synthesis of pro-inflammatory proteins that trigger an exacerbated response in the intestinal inflammatory process. Figure 9 represents a schematic suggesting the mechanisms by which NCHE attenuates the inflammatory response in the DNBS-induced intestinal inflammation model. It is possible that other components present in NCHE not identified in the present study may act synergistically, contributing to the antioxidant and anti-inflammatory effect.

## 4. Materials and Methods

### 4.1. Reagents and Plant Material

Most of the reagents in this study were obtained from Sigma-Aldrich^®^ (São Paulo, SP, Brazil) unless otherwise stated.

The cladodes of *N. cochenillifera* were harvested in May 2019 at the experimental station “Rommel Mesquita de Farias, belonging to the Empresa de Pesquisa Agropecuária do Rio Grande do Norte S/A” (EMPARN) in the Parnamirim city, Rio Grande do Norte state, Brazil (geographical coordinates: 05°55′30″ S–35°11′20″ W). The cladodes collected were young, with a dark green color and direct exposure to sunlight and without flowers or fruits. Harvest was in the rainy season.

The botanical identification voucher specimen (No. 3702) was deposited at the herbarium of the Bioscience Center of the Federal University of Rio Grande do Norte. The research was authorized by the National System for the Management of Genetic Heritage and Associated Traditional Knowledge (SISGEN process No. A5DB251). The accepted name plant was checked at http://www.worldfloraonline.org/ (accessed on 16 November 2022), possessing as a synonym *Opuntia cochinellifera* (L.) Mill. (WFO, 2022) [88].

### 4.2. Preparation of Nopalea cochenillifera Extract

The fresh cladodes were immediately proceeded after harvest. They were dried at 50 °C in a circulating air oven for 8 days and then ground in a knife mill. The raw material was extracted, by the maceration method (48 h), using as a solvent ethanol: water (70:30, *v*/*v*) in a proportion of 1:10 solvent (g/mL). The extract was filtered and the remaining raw material was extracted again for 24 h. This process was repeated 2 times, and the 3 extractions were pooled, resulting in the *N. cochenillifera* hydroethanolic extract (NCHE). The organic phase was evaporated under reduced pressure using a rotavapor (Büchi^®^) with a temperature below 40 °C and finally lyophilized and stored at 4 °C. The yield of dry extract was 4.48%.

### 4.3. Physicochemical Analysis of Nopalea cochenillifera Extract

The pH and titratable acidity were determined as recommended by AOAC (2020) [89]. pH analysis was determined using 5 g of NCHE diluted in 50 mL of distilled water with the aid of a portable electronic pH meter (KASVI^®^). Titratable acidity was determined by the potentiometric method, in a solution prepared with 1 g of the NCHE in 50 mL of distilled water, and then titrated with 0.1 M sodium hydroxide solution until the pH became basic [89]. Moisture, ash, fat, dietary fiber and protein contents were determined for NCHE according to the standard AOAC method [89]. To determine the moisture, 1.0 g of NCHE was weighed and taken to an oven with air circulation and renewal at 105 °C for 16 h (TECNAL^®^ TE-394/2-MP). Regarding the percentage of ashes, 1.0 g of NCHE was weighed in the crucible. Then, to determine the ashes percentage, crucibles were taken to the muffle furnace (Quimis Q-318M24) until reaching a temperature of 600 °C for 4 h. To determine the ether extract, extraction was performed in a Soxhlet-type apparatus. The crude fiber content of the extract was determined by the method of digesting the material in 0.255N H_2_SO_4_ solution for 40 min followed by 0.313N NaOH for another 40 min. For protein analysis, the Kjeldahl method was used, which consists of digesting the extract in sulfuric acid with a catalyst; distilling the ammonia into a receiver solution; and quantifying the ammonia by titration with a standard solution. Total carbohydrates were calculated by their difference (Equation (1)) [90].
Total carbohydrates (g/100 g) = 100 − (moisture + proteins + ether extract + ash),(1)

#### 4.3.1. Determination of Total Phenolic and Flavonoids Content

The total phenolic content (TPC) of NCHE was determined based on the Folin–Ciocalteu reagent method [91]. Briefly, in a 96-well plate, an amount of 25 μL of sample solution at 2 mg/mL was added with 125 μL of Folin–Ciocalteu reagent freshly diluted to 1:10 (*v*/*v*) with distilled H_2_O. After, an amount of 100 μL of the solution was mixed with 100 μL of a 7.5% Na_2_CO_3_ solution; the mixture was then left for 30 min in the dark, and the absorbance was measured in a microplate reader (Epoch-Biotek^®^, Winooski, Vermont, United States of America) at 765 nm. The respective standard curves of gallic acid (1.25, 2.5, 5, 10, 20, 40, 60 and 80 μg/mL) and the blank (Folin–Ciocalteu reagent with H_2_O) were constructed simultaneously. Thereafter, the TPC was calculated as mg of gallic acid equivalent (GAE) per g of the sample (mg of GAE/g of sample). The analyses were performed in triplicate in three independent experiments (three replicates, and each one was analyzed in triplicate).

Total flavonoid content (TFC) was determined by the aluminum chloride colorimetric method [92]. Sample of 50 μL of each extract (2 mg/mL) was mixed with 160 μL of ethanol (P.A.), 20 μL of aluminum chloride solution (1.8% *w*/*v*) and 20 μL of sodium acetate (8.2% *w*/*v*). The mixture was left at room temperature in the dark for 40 min, after which the absorbance was measured at 415 nm using an ELISA microplate reader after subtraction of the blank reading value and compared with a quercetin calibration curve (2.5, 5, 10, 20, 40, 60 and 80 μg/mL). The TFC was calculated as means ± SD and expressed as mg quercetin equivalent per g of the sample (mg of QE/g of sample). The analyses were performed in triplicate in three independent experiments (three replicates, and each one was analyzed in triplicate).

#### 4.3.2. Phytochemical Analysis by HPLC-ESI-MS^n^ of *Nopalea cochenillifera* Extract

HPLC-ESI-MS^n^ analysis was performed using high-performance liquid chromatography equipment (Shimadzu^®^, Kyoto, Japan), consisting of an automatic injector (Accela AS), two pumps (LC-20AD), an autosampler (SIL-20AHT) and a system controller (CBM-20A) and an Ion Trap mass spectrometer amazon-SL (Bruker Daltonics^®^), with the electrospray ionization (ESI) source operated in positive and negative mode. The chromatographic conditions were performed using a C_18_ reversed phase column (Kromasil^®^, 150 mm × 4.6 mm d.i., 5 μm) at room temperature. The sample was prepared from 1.0 mg of the NCHE in 1.0 mL of MeOH: H_2_O (50:50, *v*/*v*) and filtered using a 0.45 μm PVDF filter. The elution method adopted for NCHE was a linear gradient with a mobile phase composed of water acidified with 0.1% formic acid (eluent A) and methanol (eluent B), from 5 to 100% (B) in 60 min. The injection volume for the analysis was 20 μL and the flow rate was 600 µL/min. The MS parameters were optimized for the performance of the analysis as follows: HV End plate offset of −500 V, nebulizer of 40 psi, dry gas (N_2_) with flow of 8.0 L/min, dry temperature of 300 °C and capillary voltage of −4500 V. The acquisition range of the mass analysis was monitored from *m*/*z* 150–1200 and recorded every 0.2 s. The fragmentation experiments were performed by collision-induced dissociation (CID) running in auto MS^n^ (intelligent fragmentation) for MS^2^ and MS^3^ acquisition and with a fragmentation amplitude of 1.0 V. For precursors selection in MS^2^ and MS^3^ experiments, the most intense ions were isolated above the absolute threshold of 25,000 and 2500, respectively, and a relative intensity threshold of 5%. The data were acquired and processed using the Bruker Compass Data Analysis software (Version 4.0, Billerica, MA, United States).

For metabolite annotation, HPLC-ESI-MS^n^ data were first converted to mzML format in MSConvert software (ProteoWizard), and then MS^2^ spectra were searched against the Library Search dereplication workflow on the Global Natural Products Social Molecular Networking (GNPS) platform [93]. The matches obtained were manually verified by comparison with other databases [Dictionary of Natural Products (DNP, 2013) and LOTUS (https://lotus.naturalproducts.net/ (accessed on 16 November 2022)) [94]], as well as data previously reported in the literature. Additionally, a similar procedure was done for those spectra that did not find a match with the MS data library.

### 4.4. Animals

A total of 60 animals were included in this investigation. Female Wistar rats (200 ± 20 g, 8 weeks old) were used. Rodents were acclimated for 7 days prior to experimentation and housed under standard environment conditions at 20–25 °C and 12 h dark/light cycle and had free access to potable water (ad libitum) and standard food. The experimental protocol of the study was approved by the Committee for Ethics in Animal Experimentation of the UFRN under protocol n° 176.027/2019 in accordance with the guidelines of the National Council for the Control of Animal Experimentation (CONCEA).

### 4.5. Acute Toxicity

The acute toxicity by oral route of the NCHE was performed following the criteria recommended by the OECD, 2001 [66]. The animals were randomly divided into two groups (*n* = 6 per group). Group 1 (Control) received distilled water orally; Group 2 (acute toxicity) received a limited dose at 2000 mg/kg of the NCHE by oral route.

#### 4.5.1. Behavioral Evaluation (Hippocratic Screening and Open Field Test) and Motor (Rotarod Test)

After administration, the animals of all groups were observed with special attention in the first minutes (0, 5, 10, 15, 30, 60, 120 and 240) and 24 h after administration. The behavioral evaluation followed daily for 14 days. The analysis method applied was the Hippocratic screening, which considers the excitatory behavioral criteria, the inhibitory criteria and general activity such as defecation, diarrhea, urination, piloerection and death [95]. In addition, body mass and feed consumption were also evaluated throughout the experiment.

On days 1, 7 and 14 after the administration of the NCHE, the animals were submitted to behavioral testing using the open field apparatus in order to analyze the exploratory activity of the animals (maximum time of 5 min for each animal). On the same days (1, 7 and 14), the animals were placed on the rotating bar of the rotarod apparatus at a constant speed of 10 rpm. After 1 min of training, the time the animal remained on the bar was recorded (maximum time of 3 min) by the method described by Pereira et al., (2022) [96] with few adaptations.

#### 4.5.2. Evaluation of Hematological, Biochemical and Anatomopathological Parameters

On the 15th day, the animals were anesthetized with 10% ketamine and 2% xylazine (50 mg/kg; 10 mg/kg, intraperitoneal route), blood was collected by cardiac puncture to perform hematological and biochemical tests: glucose, triglycerides, total cholesterol (CHOL), urea, creatinine (CREA), albumin (ALB), aspartate aminotransferase (AST) and alanine aminotransferase (ALT), as analytical quality control of the analysis of lipemia and degree of hemolysis were performed. The biochemical assays were performed using standardized commercial diagnostic kits from Byosys^®^ and Kovalent^®^, in a biochemical analyzer (Achitec C8000/ABBOT). For hematological analysis: Erythrocytes, hematocrit (HCT), hemoglobin (HGB), mean corpuscular volume (MCV), mean corpuscular hemoglobin (MCH) and mean corpuscular hemoglobin concentration (MCHC), leukocytes and platelets were then determined.

The organs (liver, kidney and spleen) were collected, and their absolute weights were measured. The relative weight of these organs was obtained using the formula: organ mass/ponderal mass × 100. The organs were macroscopically analyzed, then washed with distilled water and fixed in 10% formaldehyde for 48 h. The processing of the samples was performed using the paraffin embedding technique and hematoxylin-eosin (HE) staining.

### 4.6. DNBS-Induced Colitis: Experimental Design and Treatment Protocol

Colitis induction was conducted according to Morris et al. (1989) [97], with minor modifications [98]. Female Wistar rats (*n* = 48) were randomized into six groups (*n* = 8): The healthy group (received intracolonic instillation of 0.9% normal saline only), DNBS control (were not treated after colitis induction), NCHE + colitis (received 100, 200 or 300 mg/kg for 3 days before and 2 days after colitis induction) and sulfasalazine (SSZ; received 250 mg/kg [73] for 3 days before and 2 days after colitis induction). On day 4, under light anesthesia at doses of 50 mg/kg of ketamine and 5 mg/kg of xylazine by an intraperitoneal route, and fasting overnight, animals kept in head-down positions were given 0.25 mL DNBS (25 mg dissolved in 50% (*v/v*) ethanol solution) through a Teflon cannula (2 mm diameter) inserted 8 cm into the anus. Rats from the healthy group intracolonically received 0.5 mL of 0.9% saline. The vehicle (water; control groups) or NCHE or sulfasalazine were orally administered for 12, 24 and 48 h after the colitis induction. All animals 72 h after the induction with DNBS were euthanized with anesthetic overdoses of ketamine (225 mg/kg) and xylazine (30 mg/kg) by an intraperitoneal route. The colon was extracted and photographed to evaluate the macroscopic damage. Representative samples were taken from the distal inflamed area for the histological and immunohistochemistry studies. The colonic samples were frozen at −80 °C for RT-qPCR analysis, MPO, MDA and Cytokines.

#### 4.6.1. Evaluation of Disease Activity Index (DAI), Weight/Length Ratio and Macroscopic Damage Index Colonic on DNBS Induced

The DAI was evaluated by body weight variation, the presence of rectal bleeding and stool consistency, a methodology adapted from Cooper et al. (1993) [99] and Berberat et al. (2005) [100]. At the end of treatment, the animals were euthanized, the colonic segment was weighed and their lengths were measured. After being opened longitudinally, the macroscopic damage index was evaluated according to Bell et al. (1995) [101], with scores ranging from (0) normal colon (no damage); (2) ulceration without hyperemia or wall thickening; (3) ulceration with an inflamed spot; (4) two or more sites of ulceration and/or inflammation; (5) large areas of inflammation and ulceration extending more than 1 cm; and (6–10) for large areas of tissue damage and for damage extending more than 2 cm, with 1 point added for each additional centimeter of tissue damage.

#### 4.6.2. Determination of Myeloperoxidase (MPO) Activity and Malondialdehyde (MDA) Levels in the Intestine

MPO activity was measured according to the technique described by Krawisz et al. (1984) [102]. Intestinal (colon) samples (*n* = 8) were homogenized in 0.5% hexadecyltrimethylammonium bromide (pH = 6.0; 1:20 *w*/*v*). The homogenate was centrifuged (2000× *g* at 4 °C for 20 min), and the supernatant was used to measure MPO activity. O-Dianisidine staining reagent, potassium phosphate buffer and 0.05% hydrogen peroxide at 1% were used, and the absorbance at 450 nm were subsequently determined. The results were expressed as units of MPO per gram of wet tissue; one unit of MPO activity was defined as that degrading 1 mmol of hydrogen peroxide per minute at 25 °C.

Determination of MDA content was performed by the method described by Esterbauer and Cheeseman (1990) [103]. Colonic tissue (*n* = 8) was cut with scissors into small fragments for 15 s in cold, homogenized in automatic Potter homogenizer with 20 mM Tris hydrochloride buffer (pH = 7.4; 1:5 *w*/*v*) and centrifuged at 2500× *g* at 4 °C for 10 min. The supernatants were assayed with chromogenic reagent (1-methyl-2-phenylindole 10.3 mM in 3:1 acetonitrile) and HCl (37%) for 40 min at 45 °C and centrifuged again (2500× *g* at 4 °C for 5 min). The absorbance (586 nm) was determined by a Mindray MR-96A microplate reader and calculated its content by interpolation on standard curve with 1, 1, 3,3 tetraethoxypropane (10 mM). The results were expressed as nmol/g tissue.

#### 4.6.3. Measurement of Cytokine Production in the Intestine

The intestinal tissue (*n* = 8) was cold homogenized with phosphate-buffered solution (PBS) and centrifuged (3500× *g* at 4 °C for 10 min), and the supernatant was analyzed for cytokines. The cytokine levels in the colonic tissue were evaluated using the protocols of the kits (Sigma-Aldrich, São Paulo, Brazil), using standard capture and detection antibodies for interleukin 1 beta (IL-1β), tumor necrosis factor alpha (TNF-α) and Interleukin-10 (IL-10), performed using the method described by Da Silva et al. (2018) [104].

#### 4.6.4. Analysis of RNA Transcripts by RT-qPCR

They were determined using the SV Total RNA isolation system kit (Promega Corporation^®^), following the manufacturer’s specifications. RNA was extracted from colonic tissue using the Trizol reagent (IntvitrogenTM, Waltham, MA, USA). The cDNA synthesis was performed using the High-Capacity cDNA Reverse Transcription Kit (Applied Biosystems, Waltham, MA, USA). For quantification by real-time PCR, the SYBR Green PCR Master Mix kit (Applied Biosystems) and primers (Forward-Reverse/Fw-Rv) were used, and mRNA expression was normalized using the housekeeping gene β-actin as internal control (Table 6). The mRNA relative quantification was calculated using the ΔΔCt method.

#### 4.6.5. Histopathology and Immunohistochemical Analysis

For histopathological analysis, colonic tissue samples (*n* = 8) were fixed in 4% PFA and embedded in paraffin. Subsequently, colonic sections with 3 μm thickness were obtained with a microtome and stained with hematoxylin and eosin. The histological microscopic damage index was scored on a scale from 0 to 6, following criteria pre-established by Zea-Iriarte et al. (1996) [105]. For immunohistochemistry analysis, colonic sections with 3 μm thickness were obtained with a microtome. The histological sections were deparaffinized, rehydrated, washed in 0.3% Triton X-100 in phosphate buffer and incubated with endogenous peroxidase (3% hydrogen peroxide). The sections were incubated overnight at 4 °C in the presence of primary antibodies (Santa Cruz Biotechnology): NF-κB p65 (1:100) and COX-2 (1:500). The sections were carefully rinsed with phosphate buffer and incubated in the presence of secondary antibody streptavidin/HRP-conjugated (Biocare Medical) for 30 min. Immunoreactive cells were visualized by colorimetric detection following the protocol provided by the manufacturer (TrekAvidin-HRP Label + Kit Biocare Medical). Determination of the immunostaining intensity of the cells was determined by an adaptation of the protocol described by Guerra et al. (2016) [106], and the labeled positive areas were captured by a photomicroscope (Nikon E200 LED). Histopathological and immunohistochemical analyses were performed independently by two pathologists blinded to group identity.

#### 4.6.6. Statistical Analysis

The results were expressed as the arithmetic mean ± standard deviation and the mean ± standard error of the means. Differences between means were tested for statistical significance using one-way ANOVA followed by Tukey’s test. Non-parametric data (score) were expressed as the median (range) and were analyzed using the Mann–Whitney test for histology and immunohistochemistry evaluation. All statistical analyses were performed using GraphPad 8.0.1 software (Graph-Pad Software Inc., La Jolla, CA, USA), and statistical significance was set at *p* < 0.05.

## 5. Conclusions

Overall, the present work demonstrates for the first time the safety and protective effect of a phenolic-rich extract from *N. cochenillifera* on intestinal mucosal damage in an in vivo model of induced inflammation. All in all, we offer a novel active ingredient to be applied in pharmaceuticals as a therapeutic strategy due to its potential protective effect as shown during experimental colitis by reducing colonic oxidative stress and increasing colonic antioxidant defense mechanisms and the modulation of inflammatory markers.

## Figures and Tables

**Figure 1 plants-12-00594-f001:**
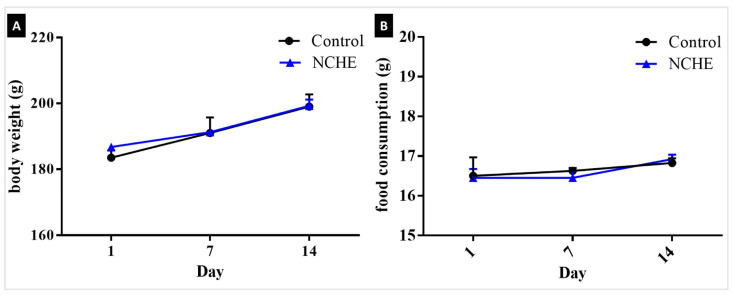
Relative body weight (**A**) and food consumption (**B**) of female rats treated with a single dose (2000 mg/kg) of *Nopalea cochenillifera* hydroethanolic extract (NCHE) observed for 14 days. Data are expressed as Mean ± SEM (*n* = 6/group). No statistical differences were detected between the treated group and the control group.

**Figure 2 plants-12-00594-f002:**
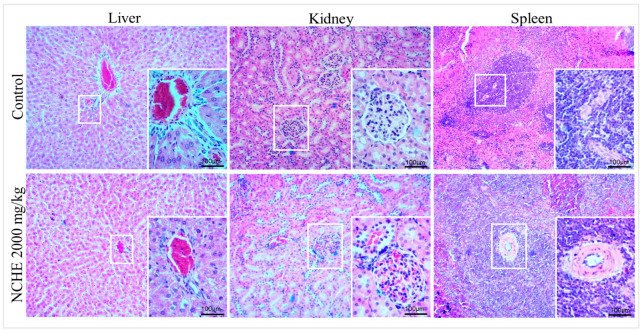
Photomicrographs of histopathological analyses of the liver, kidneys and spleen (eosin-hematoxylin stain; 10× and 40× magnification) of tissues from rats in the control group and treated with *N. cochenillifera* hydroethanolic extract (NCHE) in a single oral dose (2000 mg/kg) after 14 days.

**Figure 3 plants-12-00594-f003:**
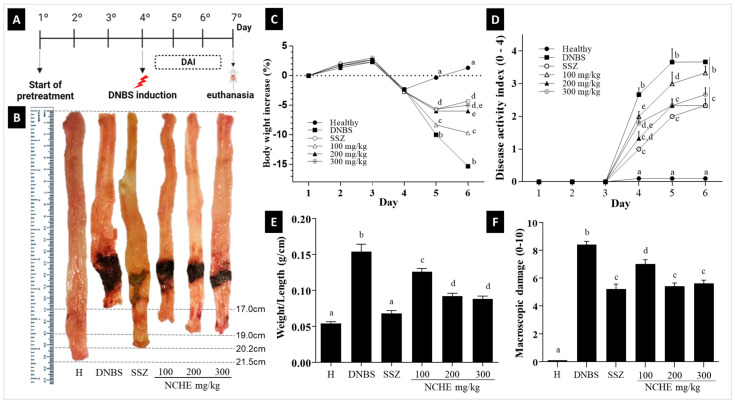
Effect of treatments with *N. cochenillifera* hydroethanolic extract (NCHE) (100, 200 and 300 mg/kg) and sulfasalazine (SSZ) (250 mg/kg) on colitis induced in rats by 2.4-Dinitrobenzene sulfonic acid (DNBS). (**A**) Experimental design; (**B**) colon damage; (**C**) weight loss; (**D**) disease activity index; (**E**) weight/length ratio; (**F**) macroscopic damage score. H: Healthy group. Data are expressed as means ± SEM. Groups with a different letter differ statistically (*p* < 0.05).

**Figure 4 plants-12-00594-f004:**
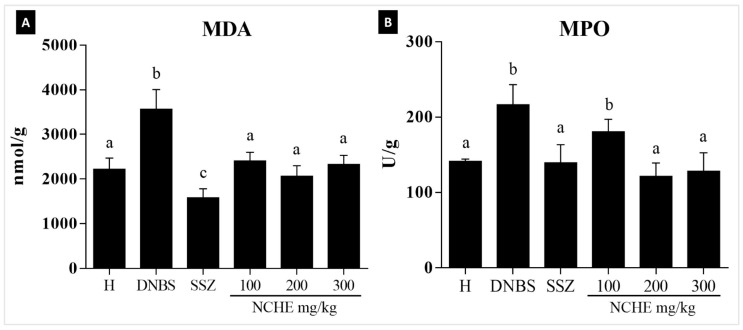
Effect of *N. cochenillifera* hydroethanolic extract (NCHE) (100, 200 and 300 mg/kg) and sulfasalazine (SSZ) (250 mg/kg) on colonic (**A**) malondialdehyde (MDA) levels and (**B**) myeloperoxidase activity. H: Healthy group. Data are expressed as means ± SEM. Groups with a different letter differ statistically (*p* < 0.05).

**Figure 5 plants-12-00594-f005:**
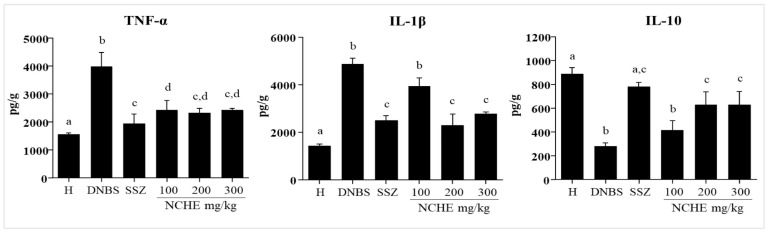
Effect of *N. cochenillifera* hydroethanolic extract (NCHE) (100, 200 and 300 mg/kg) and sulfasalazine (SSZ) (250 mg/kg) on colonic levels of the cytokines: Tumor necrosis factor alpha (TNF-α), interleukin-1beta (IL-1β), and interleukin 10 (IL-10). H: Healthy group. Data are expressed as means  ±  SEM. Groups with a different letter differ statistically (*p*  <  0.05).

**Figure 6 plants-12-00594-f006:**
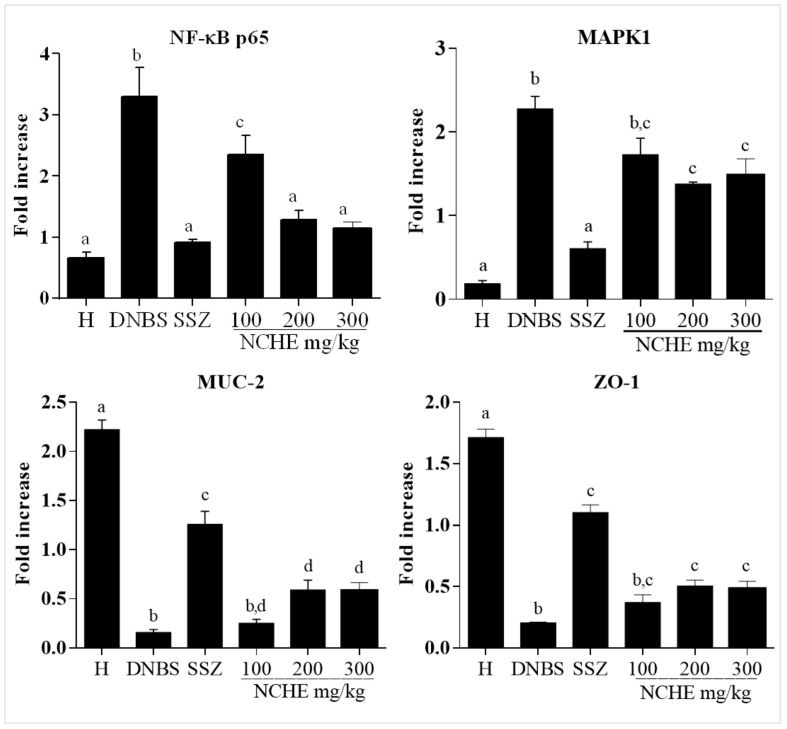
Effect of *N. cochenillifera* hydroethanolic extract (NCHE) (100, 200 and 300 mg/kg) and sulfasalazine (SSZ) (250 mg/kg) on the gene expression of mitogen-activated protein kinase 1 (MAPK1/ERK2), nuclear factor kappa B p65 (NF-κB p65), zonula occludens type I (ZO-1) and mucin type II (MUC-2) in colonic tissue of the experimental trial of acute colitis induced by 2,4-dinitrobenzene sulfonic acid (DNBS) in rats. Healthy group. Data are expressed as means ± SEM. Groups with a different letter differ statistically (*p* < 0.05).

**Figure 7 plants-12-00594-f007:**
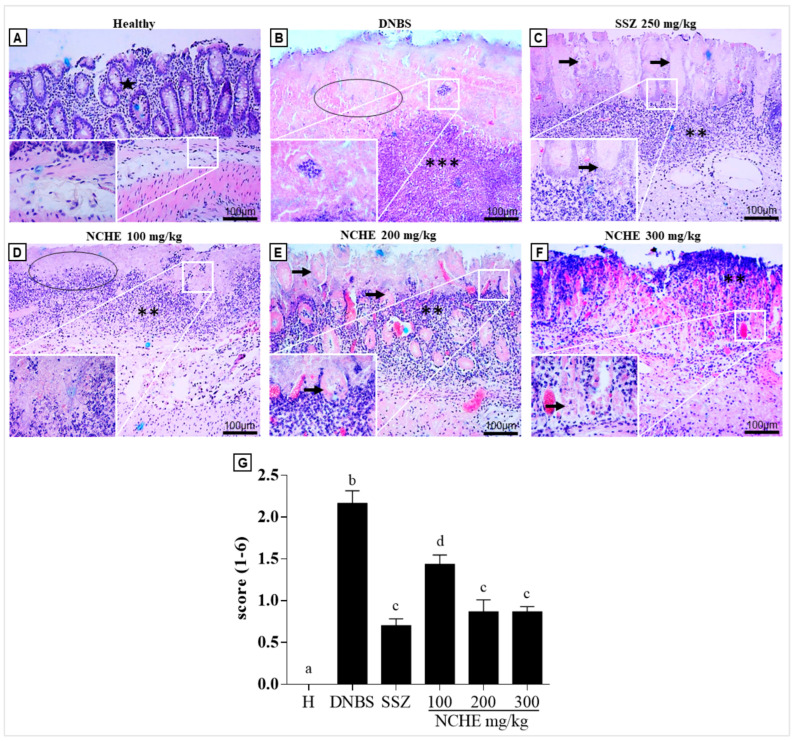
Histopathological analyses of representative colonic tissue (longitudinal section) using Hematoxylin/Eosin staining. Groups represented: H—healthy group (**A**), DNBS—DNBS control (**B**), Sulfasalazine (SSZ) (250 mg/kg) (**C**) and NCHE—*N. cochenillifera* hydroethanolic extract (100, 200 and 300 mg/kg) (**D**–**F**) and histopathological score (**G**). The figure represents microscopic damage assessment (10× and 40× magnification). Normal intestinal layers (star), inflammatory cell infiltrate intense (***) and moderate (**), total mucosal ulceration (circle), Initial mucosal regeneration (arrow). Data are expressed as means  ±  SEM. Groups with a different letter differ statistically (*p*  <  0.05).

**Figure 8 plants-12-00594-f008:**
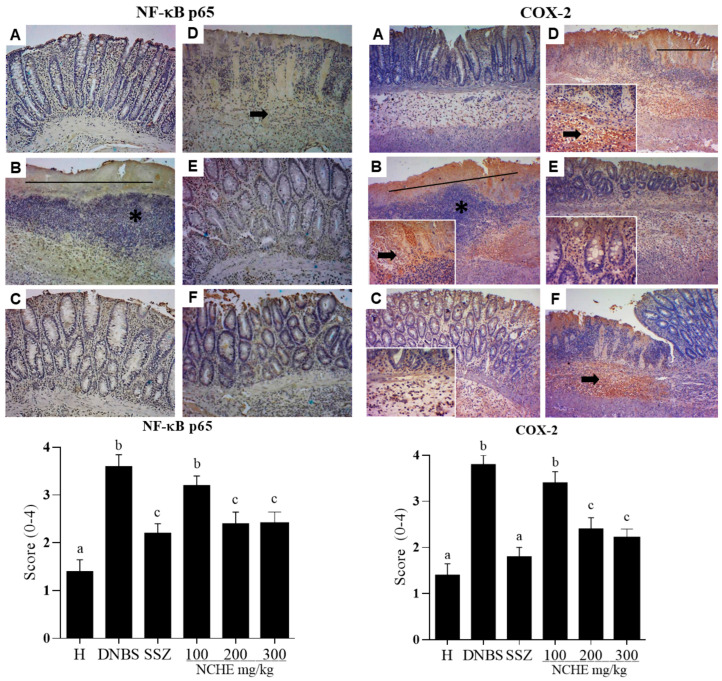
Immunohistochemical analyses of representative colonic tissue (longitudinal section). NF-κB p65 and COX-2 and their respective immunohistochemical scores. Groups represented: H—healthy group (**A**), DNBS—DNBS control (**B**), Sulfasalazine (SSZ) (250 mg/kg) (**C**) and NCHE—*N. cochenillifera* hydroethanolic extract (100, 200 and 300 mg/kg) (**D**–**F**). The figure represents a microscopic damage assessment (10× and 40× magnification), the lines indicate diffuse active colitis with intense antibody reactivity, the asterisks represent dense infiltrate of inflammatory cells and arrows indicate antibody reactivity. Data are expressed as means  ±  SEM. Groups with a different letter differ statistically (*p*  <  0.05).

**Figure 9 plants-12-00594-f009:**
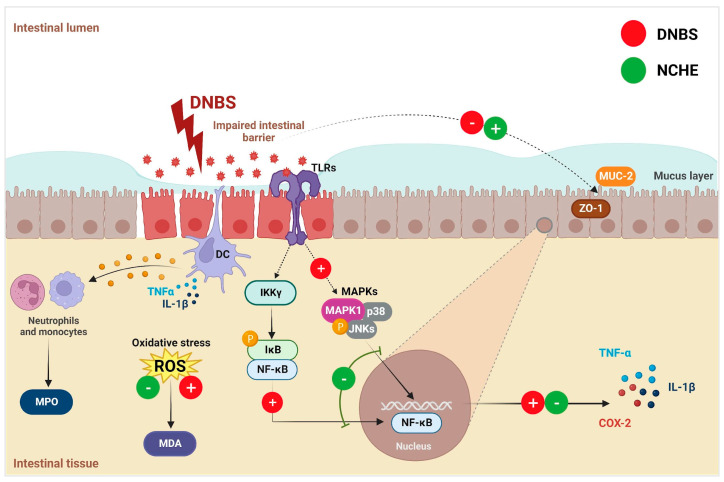
Schematic representation of the proposed mechanisms of action of dinitrobenzene sulfonic acid (DNBS) and *Nopalea cochenillifera* hydroethanolic extract (NCHE) in DNBS-induced colitis. NCHE inhibits the activation of nuclear factor kappa B (NF-κB) and mitogen activated protein kinase 1 (MAPK1) signaling pathways induced by DNBS. The suppression of the translocation of NF-κB to the nucleus can also be a possible reason for reduced levels of cyclooxygenase-2 (COX-2), interleukin-1 beta (IL-1β) and tumor necrosis factor-alpha (TNF-α). Through recognition of Pathogen Associated Molecular Patterns (PAMPs) by toll-like receptors (TLRs), gastric mucosal epithelial cells, through signal transduction pathways, can attract neutrophils and dendritic cells, favoring the production of inflammatory cytokines (IL-1β and TNF-α). It is suggested that NCHE may act by reducing neutrophil chemotaxis and neutralizing reactive oxygen species (ROS) produced by DNBS inflammation, decreasing myeloperoxidase (MPO) and malondialdehyde (MDA) levels in the tissue, respectively. NCHE can also improve DNBS-induced intestinal barrier dysfunction through positive regulation of mucin type 2 (MUC-2) and tight junction protein zonula occludens-1 (ZO-1) expression.

**Table 1 plants-12-00594-t001:** Physicochemical analysis of *N. cochenillifera* hydroethanolic extract.

Parameters	Values
pH	4.50 ± 0.04
Molar acidity (g/100 g)	4.40 ± 0.00
Moisture (g/100 g)	12.98 ± 0.51
Ashes (g/100 g)	0.002 ± 0.00
Ether extract (g/100 g)	2.78 ± 0.69
Crude fiber (g/100 g)	63.85 ± 7.32
Protein (g/100 g)	16.36 ± 0.35
Total carbohydrates (g/100 g)	67.86 ± 0.00
Total phenolics ^1^ (mg GAE/g)	67.85 ± 0.04
Total flavonoids ^2^ (mg QE/g)	46.16 ± 0.03

^1^ Gallic acid equivalent milligrams (GAE) per hundred grams of extract (mg GAE/g); ^2^ Quercetin equivalent milligrams (QE) per hundred grams of extract (mg QE/g); Data expressed as mean ± standard deviation.

**Table 2 plants-12-00594-t002:** Annotated compounds in the *N. cochenillifera* hydroethanolic extract by HPLC-ESI-MS^n^.

N.	Rt (min)	Adduct	MS (*m/z*)	MS^2^ (*m/z*)	MS^3^ (*m/z*)	Metabolite	References
**1**	4.47	[M+Cl]^−^	377	**341**	281, 179, 161	dihexose (maltose or sucrose)	[33]
**2**	4.71	[M−H]^−^	195	177, 159, **129**	101, 85, 57	gluconic acid	[34]
**3**	4.95	[M+Cl]^−^	539	**503**, 341	341, 323	trihexose (raffinose)	[35]
**4**	5.17	[M+Cl]^−^	377	**341**	281, 179, 161	dihexose (maltose or sucrose)	[33]
**5**	9.32	[M−H]^−^	191	173, **111**	67	citric acid	[36]
**6**	9.69	[M+H]^+^	135	**117**, 99	99, 71	malic acid	[37]
**7**	12.19	[M+H]^+^	284	**152**	135, 128, 110	guanosine	[38]
**8**	19.50	[M−H]^−^	447	429, 403, **315**, 297, 207, 177, 163	269, 195, 163, 153, 119	dihydroxybenzoic acid *O*-pentose-hexose	[39]
**9**	21.42	[M−H]^−^	447	**315**	153, 123	dihydroxybenzoic acid *O*-pentose-hexose	[39]
**10**	22.46	[M+H]^+^	195	**177**	163, 145, 117	ferulic acid	[40]
**11**	24.36	[M−H]^−^	355	295, 265, 235, 217, **193**, 175, 160, 134	178, 149, 134	ferulic acid *O*-hexose	[40]
**12**	25.75	[M−H]^−^	355	217, **193**, 175	178, 149, 134	ferulic acid *O*-hexose	[40]
**13**	27.53	[M+H]^+^	169	**137**	109, 93, 81	vanillic acid	[41]
**14**	30.98	[M+H]^+^	757	611, 465, **303**	285, 257, 229, 165, 153	quercetin-*O*-hexose-deoxyhexose	[42]
**15**	31.89	[M+H]^+^	476	**314**	177, 145	feruloyl tyramine-*O*- hexose	[43]
**16**	32.98	[M+H]^+^	771	625, 479, 463, 427, **317**, 302	302, 285, 274, 257, 153	methylquercetin-*O*- hexose-deoxyhexose-deoxyhexose	[42]
**17**	34.32	[M+H]^+^	562	386, **314**	177, 145	feruloyl tyramine-*O*- (malonyl)-hexose	[44]
**18**	35.19	[M+H]^+^	611	593, 465, **303**	285, 257, 229, 165	quercetin-*O*-hexose-deoxyhexose (rutin)	[45]
**19**	36.56	[M−H]^−^	445	**283**	268	hydroxy-methoxy- isoflavone-*O*-hexose	[42]
**20**	37.27	[M+H]^+^	461	**299**	284, 266	hydroxy-dimethoxy- isoflavone-*O*-hexose	[42]
**21**	37.46	[M+H]^+^	314	**177**, 145, 117	145, 117	feruloyltyramine	[44]
**22**	38.15	[M+H]^+^	595	449, **287**	269, 258, 241, 231, 213, 197, 165, 153, 121	kaempferol-*O*-hexose-deoxyhexose	[42]
**23**	38.60	[M+H]^+^	625	479, **317**, 302	302, 285, 229, 165, 153	methylquercetin-*O*-hexose-deoxyhexose	[42]
**24**	38.83	[M+H]^+^	547	**299**, 284, 266	284, 266	hydroxydimethoxy-isoflavone-*O*-(malonyl)-hexose	[42]
**25**	39.91	[M+H]^+^	547	**299**, 284, 266	284, 266, 239	hydroxydimethoxy-isoflavone-*O*-(malonyl)-hexose	[42]

In bold are the precursor ions of MS^n^ experiments.

**Table 3 plants-12-00594-t003:** Biochemical and hematological parameters of female rats treated with a single oral dose (2000 mg/kg) of *Nopalea cochenillifera* hydroethanolic extract (NCHE).

Biochemical Parameters			
Parameters	Control^ns^	NCHE^ns^ (2000 mg/kg)	Reference Wistar (Females)
Glucose (mg/dL)	159 ± 5.39	151 ± 9.26	53–172
Triglycerides (mg/dL)	48 ± 9.91	54 ± 5.26	23–138
Total Cholesterol (mg/dL)	69 ± 5.55	62 ± 3.33	54–96
Urea (mg/dL)	36 ± 5.55	33 ± 5.26	24–49
Creatinine (mg/dL)	0.6 ± 0.05	0.5 ± 0.08	0.3–1.1
Albumin (mg/dL)	2.9 ± 0.09	3.0 ± 0.04	1.3–3.8
AST (U/L)	99 ± 2.90	86 ±8.53	51–211
ALT (U/L)	50 ± 0.69	52 ± 0.55	32–62
**Hematological Parameters**			
Erythrocytes (×10^6^/µL)	7.53 ± 5.39	7.28 ± 9.26	5.21–8.83
Hemoglobin (g/dL)	13.7 ± 2.20	13.40 ± 1.13	11.1–17.10
Hematocrit (%)	44 ± 2.95	43.00 ± 2.43	27.00–49.00
MCV (fL)	51.27 ± 1.55	49.86 ± 4.36	45.00–56.70
MCH (pg)	18.41 ± 1.95	20.06 ± 1.06	16.60–22.80
MCHC (g/dL)	30.74 ± 0.12	31.35 ± 0.98	30.40–43.90
Leukocytes (cell/µL)	6400 ± 1920	5900 ± 1150	2300–9900
Platelets (×10^6^/µL)	904 ± 96	1.025 ± 89	760–1.310

AST: aspartate aminotransferase, ALT: alanine aminotransferase, MCV: Mean Corpuscular Volume, MCHC: Mean Corpuscular Hemoglobin, MCHC: Mean Corpuscular Hemoglobin Concentration. ns: not significant. Data are expressed as Mean ± SEM (*n* = 6/group). No statistical differences were detected between the treated group and the control group.

**Table 4 plants-12-00594-t004:** Behavioral evaluation in the open field test and motor activity evaluation in the rotarod test in rats on days 1, 7 and 14 post-treatment with *N. cochenillifera* hydroethanolic extract (NCHE) in a single oral dose (2000 mg/kg).

Open Field Test			
Parameters	Day	Control	NCHE (2000 mg/kg)
Total distance traveled (cm)	1	230.80 ± 3.93	250.20 ± 5.54
	7	240.20 ± 2.15	220.20 ± 4.76
	14	260.80 ± 3.50	250.80 ± 4.52
Rearing or climbing behavior (count)	1	14.20 ± 2.25	16.40 ± 8.40
	7	12.32 ± 5.28	12.20 ± 4.38
	14	16.00 ± 6.06	19.6 ± 4.76
Grooming (count)	1	8.60 ± 1.95	9.60 ± 2.91
	7	7.00 ± 2.80	8.74 ± 2.67
	14	7.65 ± 1.54	9.80 ± 2.96
Number of defecations	1	1.00 ± 1.00 ^a^	3.80 ± 1.00 ^b^
	7	1.60 ± 1.02 ^a^	3.80 ± 0.98 ^b^
	14	0.80 ± 0.53 ^a^	4.20 ± 2.00 ^b^
**Rotarod Test**			
Staying Time (seconds)	1	180.0 ± 0.00	179.6 ± 0.40
	7	180.0 ± 0.00	179.0 ± 1.00
	14	180.0 ± 0.00	180.0 ± 0.00

Values represent the mean ± SEM (*n* = 6/group). Different letters differ statistically.

**Table 5 plants-12-00594-t005:** Relative weight of the organs of the rats in the control group and treated with *N. cochenillifera* hydroethanolic extract (NCHE) in a single oral dose (2000 mg/kg).

Relative Organ Weight (g/100 g Body Weight)		
Organ	Control^ns^	NCHE^ns^ (2000 mg/kg)^ns^
Liver	3.80 ± 0.12	4.10 ± 0.02
Spleen	0.28 ± 0.03	0.30 ± 0.04
Kidneys	0.90 ± 0.03	0.87 ± 0.04
Heart	0.35 ± 0.01	0.37 ± 0.02
Lung	0.42 ± 0.09	0.42 ± 0.20

ns: not significant. Data are expressed as Mean ± SEM (*n* = 6/group). No statistical differences were detected between the treated group and the control group.

**Table 6 plants-12-00594-t006:** Primer sequences used in real-time PCR assays. Beta-actin (β-actin), mitogen-activated protein kinase 1 (MAPK1), nuclear factor kappa B p65 (NF-κB p65), zonula Occludens type I (ZO-1), mucin type II (MUC-2).

Gene	Sequence 5′–3′	Annealing Temperature (°C)
*β-actin*	Fw—CGCACTGCCGCATCCTCT	58
	Rv—GTCGAAGAGAGCCTCGG	
*MAPK1*	Fw—CCCAAGTGATGAGCCCATTG	58
	Rv—GGTAAGTCGTCCAGCTCCATGT	
*NF-κB p65*	Fw—TCTGCTTCCAGGTGACAGTG	58
	Rv—ATCTTGAGCTCGGCAGTGTT	
*ZO-1*	Fw—GGGGCCTACACTGATCAAGA	56
	Rv—TGGAGATGAGGCTTCTGCTT	
*MUC-2*	Fw—GCAGTCCTCAGTGGCACCTC	60
	Rv—CACCGTGGGGCTACTGGAGAG	

## Data Availability

Not applicable.

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
