# Peer review of "Toxicity and Anti-Inflammatory Activity of Phenolic-Rich Extract from Nopalea cochenillifera (Cactaceae): A Preclinical Study on the Prevention of Inflammatory Bowel Diseases"

_plants, 2023, doi:10.3390/plants12030594_

Round 1

Reviewer 1 Report

The manuscript entitled " Toxicity assessment and anti-inflammatory activity of phenolic- rich extract from Nopal cactus (Nopalea cochenillifera): A preclinical study focusing on the prevention of inflammatory bowel diseases" presented by Tavares et al, summaries a research study towards establishment role of Nopalea cochenillifera towards prevention of inflammatory bowel diseases by experimental finding. Overall, i found manuscript interesting but there are some scientific flaws need to be addressed in current forms. Some of my observations are:    

·       Tile must be concise

·       In abstract, “The dose of 2000 mg/kg by oral route showed no 32 signs of toxicity, mortality or significant changes in biochemical and hematological parameters”. Please mention name of extract.

·       In introduction please provide literature related to earlier pharmacological finding on plant. Also how and why this study is planned…is there any past lead

·       Instead of using hydroethanolic..please use hydro-alcohol.

·       Why any positive control group not utilized as reference standard during in vivo study.

·       Is there any past study related to phenolic composition and its role in IBD must be discussed with your present findings.

·       Language and any other typological mistake can be address

Author Response

Enclosed you will find our answers/comments to the reviewers. The paper was revised according to referee suggestions and new material was included as is possible. All suggestions were accepted, answered or justified.

We will be looking forward to hearing from Plants, soon.

Sincerely,

Silvana Zucolotto

The manuscript entitled "Toxicity assessment and anti-inflammatory activity of phenolic- rich extract from Nopal cactus (Nopalea cochenillifera): A preclinical study focusing on the prevention of inflammatory bowel diseases" presented by Tavares et al, summaries a research study towards establishment role of Nopalea cochenillifera towards prevention of inflammatory bowel diseases by experimental finding. Overall, I found manuscript interesting but there are some scientific flaws need to be addressed in current forms. Some of my observations are:   

Q1. Title must be concise

Reply to reviewer:  First, we would like to thank the reviewers for their suggestions. The title was revised.

Q2. In abstract, “The dose of 2000 mg/kg by oral route showed no 32 signs of toxicity, mortality or significant changes in biochemical and hematological parameters”. Please mention name of extract.

Reply to reviewer:  We appreciate the reviewer’s comment. The name of extract was added in abstract.

Q3. In introduction please provide literature related to earlier pharmacological finding on plant. Also how and why this study is planned…is there any past lead

Reply to reviewer: Thanks for the comment. The literature reports few pharmacological studies with this species. The following paragraph was added in the introduction:

Previous studies have reported that this plant showed potential antibiotic and antifungal activities in vitro assays [24,25] and decreased the blood glucose level in vivo study [31] and in a pilot clinical trial [30]. Regarding anti-inflammatory effect, the oral administration of the hydroethanolic extract of N. cochenillifera cladodes showed significant an-ti-inflammatory activity in the rat granuloma induction method [30].

Q4.  Instead of using hydroethanolic. Please use hydro-alcohol.

Reply to reviewer:  If you agree we prefer to keep the term hydroethanolic extract. We had published other papers with this term.

Q5.    Why any positive control group not utilized as reference standard during in vivo study.

Reply to reviewer: In the experimental design, the control group Sulfasalazine (SSZ) was used as a reference standard during in vivo study. In addition we used “healthy control group” and “colitic control group”. Follow the distribution of the groups:

Female Wistar rats (n = 48) were randomized into six groups (n = 8):

1- Healthy group – No-colitic (received intracolonic instillation of 0.9% normal saline only),

2- DNBS-control (were not treated after colitis induction),

3-NCHE + colitis received 100 mg/kg for 3 days before and 2 days after colitis induction)

4- NCHE + colitis received 200 mg/kg for 3 days before and 2 days after colitis induction)

5- NCHE + colitis received 300 mg/kg for 3 days before and 2 days after colitis induction)

6- Colitis + sulfasalazine (SSZ; received 250 mg/kg for 3 days before and 2 days after colitis induction).

Q6.    Is there any past study related to phenolic composition and its role in IBD must be discussed with your present findings.

Reply to reviewer:

Several scientific studies have confirmed that polyphenols have protective and therapeutic effects in the management of IBD mediated by downregulation of cytokines and inflammatory enzymes, enhancing antioxidant defense, and suppressing inflammatory pathways and their cell signaling mechanisms (FARZAEI et al., 2015; SHAPIRO et al., 2007; ARYA et al., 2020).

The following paragraph is present in the discussion of our study:

Clinical studies on flavonoid supplementation in IBD patients have shown a positive correlation in maintaining remissive disease state (GLABSKA et al., 2017; SKOLMOWSKA et al, 2019). Another clinical study conducted with the polyphenol curcumin in combination with mesalazine also showed promising results, patients in the curcumin group reported symptomatic improvements of fewer bowel movements, less diarrhea, less abdominal pain and cramps, endoscopic remission and an overall positive clinical response (BANERJEE et al., 2021).

Q7. Language and any other typological mistake can be address

Reply to reviewer: The entire manuscript language was revised.

Reviewer 2 Report

The manuscript is very well written and all the required experiments are done and explained in chronological order. 

  I have only one concern if the author tried to study pure compounds versus the crude mixture. especially when authors are talking about preclinical studies. It would be great if authors can include the study of pure compounds as well. 

Author Response

Dear Reviewer,

Enclosed you will find our answers/comments to the reviewers. The paper was revised according to referee suggestions and new material was included as is possible. All suggestions were accepted, answered or justified.

We will be looking forward to hearing from Plants, soon.

Sincerely,

Silvana Zucolotto

Q1. The manuscript is very well written and all the required experiments are done and explained in chronological order.

Reply to reviewer: Thanks for the comment

Q2. I have only one concern if the author tried to study pure compounds versus the crude mixture. Especially when authors are talking about preclinical studies. It would be great if authors can include the study of pure compounds as well.

Reply to reviewer: Thanks for the comment and we agree with you that it is so important to investigate the mechanism of action of some main compounds identified in the extract, but it is our first publication with this species and a dereplication study through HPLC-ESI-MSn  was included. The goal of our work is to develop a novel raw material with N. cochenilliefera crude extract instead to obtain new isolated compounds. In the further study our idea is to isolate and identify the main compound in a large amount enough for pharmacological e quality control studies (to define its content in dry extract through HPLC).

Reviewer 3 Report

Toxicity assessment and anti-inflammatory activity of phe- 2 nolic- rich extract from Nopal cactus (Nopalea cochenillifera): A 3 preclinical study focusing on the prevention of inflammatory 4 bowel diseases

The present study investigates the phytochemical profile of N. cochenillifera extract, and evaluates its acute toxicity and anti-inflammatory effect on 2,4-dinitroben-zenesulfonic acid (DNBS)-induced colitis in rats. The authors should address the following concerns:

·         Authors should revise the introduction by including the findings of previous studies which demonstrate the anti-inflammatory property of Nopal cactus

·         Include the reference for the dose of sulfasalazine used in the study

·         How did the authors suspend the hydroalcoholic extract of Nopal cactus for oral delivery in mice?

·         The results of ‘Number of defecation’ in Table 4 showed a significant increase in NCHE (2000 mg/kg) treated animals compared to control. What can be the cause for that much increase in the extract-supplemented group?

·         Figure legend of Table 5 is showing discrepancies with the data presented.

·         Statistical analysis needs rechecking.

·         2.3.3: It is stated that “TNF-α levels were significantly increased in the healthy group (p < 0.001), SSZ group 205 (p < 0.01) and in all groups treated with NCHE, compared to the DNBS group” The results in the table 5 shows opposite results.

·         2.3.3: It is stated that “There was no significant difference in IL-1β levels between the NCHE (200 mg/kg) and DNBS group”. The results in the table 5 shows significant difference (p < 0.001) between DNBS group and NCHE (200 mg/kg).

·         Discussion is too lengthy. Revise the discussion by giving emphasis to the present study results by citing previously reported studies.

·         Statistical analysis needs rechecking.

·          The weakness that dampened the enthusiasm was a general lack of clarity in the overall presentation of the ideas and a lack of logical flow of the sentence structures and ideas/results, which falls behind in an overall good scientific writing. Please have the manuscript proofread by all contributors. There are grammatical and typographic errors at multiple instances.

Author Response

Dear Reviewer,

Enclosed you will find our answers/comments to the reviewers. The paper was revised according to referee suggestions and new material was included as is possible. All suggestions were accepted, answered or justified.

We will be looking forward to hearing from Plants, soon.

Sincerely,

Silvana Zucolotto

The present study investigates the phytochemical profile of N. cochenillifera extract, and evaluates its acute toxicity and anti-inflammatory effect on 2,4-dinitroben-zenesulfonic acid (DNBS)-induced colitis in rats. The authors should address the following concerns:

      Q1. Authors should revise the introduction by including the findings of previous studies which demonstrate the anti-inflammatory property of Nopal cactus.

Reply to reviewer: Thanks for the comment.

NECCHI et al., 2011, investigated the anti-inflammatory activity of 70 % ethanolic extract of Nopalea cochenillifera in granulomatous tissue induction model and kidney and liver toxicity by serum dosage in rats.

The literature reports few pharmacological studies with this species. The following sentence was added in the introduction:

Previous studies have reported that this plant showed potential antibiotic and antifungal activities in vitro assays [24,25] and decreased the blood glucose level in vivo study [31] and in a pilot clinical trial [30]. Regarding anti-inflammatory effect, the oral administration of the hydroethanolic extract of N. cochenillifera cladodes showed significant anti-inflammatory activity in the rat granuloma induction method [30].

Q2.  Include the reference for the dose of sulfasalazine used in the study

Reply to reviewer: Thanks for the comment. The reference for the dose of sulfasalazine used in the study was included in the manuscript: Andrade at al., 2020. Anti-Inflammatory and Chemopreventive Effects of Bryophyllum pinnatum (Lamarck) Leaf Extract in Experimental Colitis Models in Rodents. https://doi.org/10.3389/fphar.2020.00998

Q3. How did the authors suspend the hydroalcoholic extract of Nopal cactus for oral delivery in mice?

Reply to reviewer: In this study, the vehicle used to solubilize the extract and sulfasalazine was water.

Q4. The results of ‘Number of defecation’ in Table 4 showed a significant increase in NCHE (2000 mg/kg) treated animals compared to control. What can be the cause for that much increase in the extract-supplemented group?

Reply to reviewer: In the present experiment, different patterns of defecation were observed between controls and rats treated with the extract. Initially it seems to have been due to the fiber content present in the extract, but this effect was maintained during 14 days. Other variables related to anxiety and motor behavior did not show significant differences, so we cannot assign this isolated parameter as an indicator of toxicity.

Q6. Figure legend of Table 5 is showing discrepancies with the data presented.

Reply to reviewer: We apologize for the mistake. The Figure legend of Table 5 has been modified.  

Q7. Statistical analysis needs rechecking.

Reply to reviewer: We appreciate the reviewer’s comment. The statistical analysis was revised.

Q8. 2.3.3: It is stated that “TNF-α levels were significantly increased in the healthy group (p < 0.001), SSZ group 205 (p < 0.01) and in all groups treated with NCHE, compared to the DNBS group” The results in the table 5 shows opposite results.

Reply to reviewer: Thanks for the comment. The text has been changed to: “TNF-α levels were significantly increased in the DNBS-control group (p < 0.05) and remained at baseline levels in the healthy group (p < 0.05). On the other hand, it reduced in the SSZ group (p < 0.05) and in all groups treated with NCHE, compared to the DNBS group.

Q9. 2.3.3: It is stated that “There was no significant difference in IL-1β levels between the NCHE (200 mg/kg) and DNBS group”. The results in the table 5 shows significant difference (p < 0.001) between DNBS group and NCHE (200 mg/kg).

Reply to reviewer: Thanks for the comment. The text has been changed to: “Additionally, the pre-treatment revealed that doses of NCHE at 200 and 300 mg/kg significantly reduced IL-1β (p < 0.05) levels. However, NCHE 100 mg/kg did not differ statistically (p > 0.05) from the DNBS control.”

Q10. Discussion is too lengthy. Revise the discussion by giving emphasis to the present study results by citing previously reported studies.

Reply to reviewer: Thanks for the comment. The discussion was revised and improved.

Q11. Statistical analysis needs rechecking.

Reply to reviewer: We appreciate the reviewer’s comment. The statistical analysis was revised.

Q12. The weakness that dampened the enthusiasm was a general lack of clarity in the overall presentation of the ideas and a lack of logical flow of the sentence structures and ideas/results, which falls behind in an overall good scientific writing. Please have the manuscript proofread by all contributors. There are grammatical and typographic errors at multiple instances.

Reply to reviewer: An extensive review was conducted in the discussion of the manuscript. We included more information about the state of the art of Nopalea cochenillifera and linked the paragraphs to achieve a logical flow.